# Fitness-Conditional Genes for Soil Adaptation in the Bioaugmentation Agent *Pseudomonas veronii* 1YdBTEX2

Marian Morales,[a] Vladimir Sentchilo,[a] Nicolas Carraro,[a] Senka Causevic,[a] Dominique Vuarambon,[a] Jan R. van der Meer[a]

[a]Department of Fundamental Microbiology, University of Lausanne, Lausanne, Switzerland

**ABSTRACT** Strain inoculation (bioaugmentation) is a potentially useful technology to provide microbiomes with new functionalities. However, there is limited understanding of the genetic factors contributing to successful establishment of inoculants. This work aimed to characterize the genes implicated in proliferation of the monoaromatic compound-degrading *Pseudomonas veronii* 1YdBTEX2 in nonsterile polluted soils. We generated two independent mutant libraries by random minitransposon-delivered marker insertion followed by deep sequencing (Tn-seq) with a total of $5.0 \times 10^5$ unique insertions. Libraries were grown in multiple successive cycles for up to 50 generations either in batch liquid medium or in two types of soil microcosms with different resident microbial content (sand or silt) in the presence of toluene. Analysis of gene insertion abundances at different time points (passed generations of metapopulation growth), in comparison to proportions at start and to *in silico* generated randomized insertion distributions, allowed to define ~800 essential genes common to both libraries and ~2,700 genes with conditional fitness effects in either liquid or soil (195 of which resulted in fitness gain). Conditional fitness genes largely overlapped among all growth conditions but affected approximately twice as many functions in liquid than in soil. This indicates soil to be a more promiscuous environment for mutant growth, probably because of additional nutrient availability. Commonly depleted genes covered a wide range of biological functions and metabolic pathways, such as inorganic ion transport, fatty acid metabolism, amino acid biosynthesis, or nucleotide and cofactor metabolism. Only sparse gene sets were uncovered whose insertion caused fitness decrease exclusive for soils, which were different between silt and sand. Despite detectable higher resident bacteria and potential protist predatory counts in silt, we were, therefore, unable to detect any immediately obvious candidate genes affecting *P. veronii* biological competitiveness. In contrast to liquid growth conditions, mutants inactivating flagella biosynthesis and motility consistently gained strong fitness advantage in soils and displayed higher growth rates than wild type. In conclusion, although many gene functions were found to be important for growth in soils, most of these are not specific as they affect growth in liquid minimal medium more in general. This indicates that *P. veronii* does not need major metabolic reprogramming for proliferation in soil with accessible carbon and generally favorable growth conditions.

**IMPORTANCE** Restoring damaged microbiomes is still a formidable challenge. Classical widely adopted approaches consist of augmenting communities with pure or mixed cultures in the hope that these display their intended selected properties under *in situ* conditions. Ecological theory, however, dictates that introduction of a nonresident microbe is unlikely to lead to its successful proliferation in a foreign system such as a soil microbiome. In an effort to study this systematically, we used random transposon insertion scanning to identify genes and possibly, metabolic subsystems, that are crucial for growth and survival of a bacterial inoculant (*Pseudomonas veronii*) for targeted degradation of monoaromatic compounds in contaminated nonsterile soils. Our results indicate that although many gene functions are important for proliferation in soil, they are general factors for growth and not exclusive for soil. In other words, *P. veronii* is a generalist

Address correspondence to Jan R. van der Meer, Janroelof.vandermeer@unil.ch.

The authors declare no conflict of interest.

that is not *a priori* hindered by the soil for its proliferation and would make a good bio-augmentation candidate.

**KEYWORDS** bioremediation, soil microbiome, toluene degradation, transposon insertion sequencing

Bioaugmentation to accelerate or extend pollutant degradation rates is a potentially useful biotechnology for cleanup of contaminated sites (1, 2), but relies on ecological properties of inoculant bacteria (and resident microbiota) that are still generally poorly understood. Implicit to many efforts is the hypothesis that wild-type strains isolated from polluted soils may have the advantage that efficient degradation capacity is coupled to some selected inherent adaptability to stressful conditions. This may favor their functioning in deteriorated environments with similar pollution conditions. In practice, however, strain inoculation only shows moderate success and inoculants frequently grow and survive poorly in soils upon reinoculation (3–7). There is thus a need to turn from trial-and-error to more systematic approaches and characterize the functions and capacities of strains to optimally respond and adapt to their new growth environment (8), and to understand whether different bacteria have different strategies in this respect.

Past studies have highlighted a variety of mechanisms important for bacterial proliferation in soils, including osmoprotectant strategies to cope with changing water potential (9–13), the expression of multiple efflux systems capable of extruding heavy metals or toxic solvent molecules (8, 14, 15), the presence of alternate respiratory pathways to survive under microaerophilic or anaerobic conditions (16–18), or the capacity of siderophore production to thrive under low iron conditions (19, 20). In addition, contaminant degradation itself may require adaptative defenses to modify membranes and cell surface structures (21–23), or to counteract generated reactive oxygen species (24–27). Many of these mechanisms have been captured by using global transcriptomics to follow the complex multifactorial physiological and metabolic responses of inoculants during a transition into contaminated environmental compartments (10, 28–32). Comparative transcriptomic studies further suggested that soil survival and growth may require specific adaptation programs that seemingly have little overlap between different bacteria (33), but their importance has not been further investigated.

Inferring the global importance of gene functions to environmental survival can be facilitated by transposon insertion sequencing. Transposon sequencing is based on the activity of transposases to insert DNA fragments with recognizable ends (34, 35). Whereas natural transposon activity is subject to extensive regulatory control by the host cell, synthetic hyperactive transposon constructs allow the production of bacterial libraries, in which each member is a unique mutant generated by the random insertion of a marker gene delivered by that transposon. This massive transposon mutagenesis can be coupled with deep sequencing methods to identify and quantify all insertions *en masse* in a mutant library (hence, also known as Tn-seq) (36–41). The basic premise in Tn-seq is that if a marker insertion has inactivated a gene, and the gene function is essential or has a fitness effect, the proportion of the mutant in the overall library will change over time with respect to the original library, or even disappear from the metapopulation of all mutants. Problematic for many of the Tn-seq applications is that the library preparation itself requires selection for the marker insertion and thus some growth, resulting in insertions in essential genes being counterselected and absent at the start (42). Tn-seq has been widely utilized to study specific metabolic pathways (43–45), gene essentiality (38, 41, 42), or mechanisms of pathogenesis (46, 47), but has only rarely been used to address the question of inoculant environmental survival (48, 49).

The major objectives of the underlying study were to understand the specific functions and metabolic subprograms that contribute to environmental survival and growth in soil of a candidate inoculant for targeted remediation of monoaromatic compounds. To test this, we used the bacterium *Pseudomonas veronii* 1YdBTEX2, which grows on monoaromatic compounds as a sole carbon and energy source (50–52). The

strain was naturally enriched in jet fuel-contaminated soils (53), and thus might have been selected for its soil resistance or survival capacity. We have previously studied the global reactions of *P. veronii* 1YdBTEX2 to transitions from laboratory to soil (52, 54) and showed that the strain can proliferate in contaminated nonsterile soils at microcosm scales (54). We further developed a genome-scale model to better understand its metabolic reprogramming during growth transitions (55).

To detect *P. veronii* genes specifically determining fitness in soil, we generated two independent high-density random Tn-seq insertion libraries. These libraries were inoculated and cultured in four independent replicates for up to 15 growth-dilution cycles in two different nonsterile soils (sand and silt) supplemented with toluene to allow *P. veronii* growth and survival. This was contrasted to growth-dilution series of the same mutant libraries in liquid suspension with toluene, to tease out effects originating from selection for toluene growth in minimal medium from those of growing and surviving in soil. Soils were nonsterile and contained different microbial community backgrounds, which were characterized by 16S rRNA V3-V4 variable gene region amplicon sequencing (for prokaryotes) and metatranscriptomic analysis (for eukaryotes). Cell densities of the *P. veronii* mutant metapopulation in soil microcosms were determined after each cycle, and total DNA was extracted at three points of increasing estimated generations of metapopulation growth. Positions and abundances of insertions were determined by Illumina high-throughput sequencing, compared over time and as a function of treatment, and further compared to randomized *in silico* insertion models to estimate fitness effects. Groups of genes affecting fitness were characterized based on their annotation and biological function, and by means of metabolic pathway or clustered orthologies. One *P. veronii* mutant whose insertion suggested strong fitness gain in soil was reproduced by a clean individual deletion and reexamined for growth in nonsterile silt microcosms. Our results indicate highly selective and reproducible fitness increase in soils of mutants with gene defects in flagellar motor complex synthesis, but relatively sparse common traits with fitness decrease specific to soils. In contrast, gene insertions with fitness decrease in both soil conditions largely overlap those found in liquid growth. Growth in liquid leads to disappearance of twice as many gene insertions, indicating that soil is a more permissive growth environment for *P. veronii* and supports maintenance of many mutants with conditional fitness defects, possibly through traces of available nutrients or community nutrient sharing.

## RESULTS

**Genome-wide random insertion libraries of *P. veronii*.** Two independent random insertion libraries in *P. veronii* 1YdBTEX2 (Lib1 and Lib2) were produced by conjugating a nonreplicating plasmid encoding a hyperactive Tn*5* transposon from an *E. coli* donor (56). The transposon inserts an *aph* marker gene (Km-resistance) flanked by Tn*5*-ends, which is used to counterselect wild-type recipient cells. *E. coli* donor cells were depleted by phage treatment. Sequencing of genomic inserts from the Tn*5*-left end (Tn-seq) revealed 303,756 and 244,542 unique insertion positions in *P. veronii* Lib1 and Lib2 libraries, respectively, and 500,694 unique insertions combined among them (Fig. 1A). This corresponded to a mean starting density of 156 to 200 unique insertions per 10 kb (median of 16 per 10 kb; Fig. 1B). The libraries had few (47,604) overlaps, indicating the mostly independent random targets of the transposon insertions on the genome. $Log_2$ per 10 kb insertion density distributions were similar among both libraries, but the variation still suggested a considerable difference in transposed marker target selectivity or implicit fitness selection during library preparation and propagation (Fig. 1B). Insertion densities in Chromosome 1 (Chr1) were significantly lower than the other two *P. veronii* replicons (Chromosome 2 [Chr2] and Plasmid [Plm]; Fig. 1C and D), probably because more genes with essential functions are located on Chr1. A slight "smiley" effect of insertion densities in Chromosome 1 was visible, which is likely the result of chromosome replication effects (i.e., more target DNA being available close to the origin of replication; Fig. 1A).

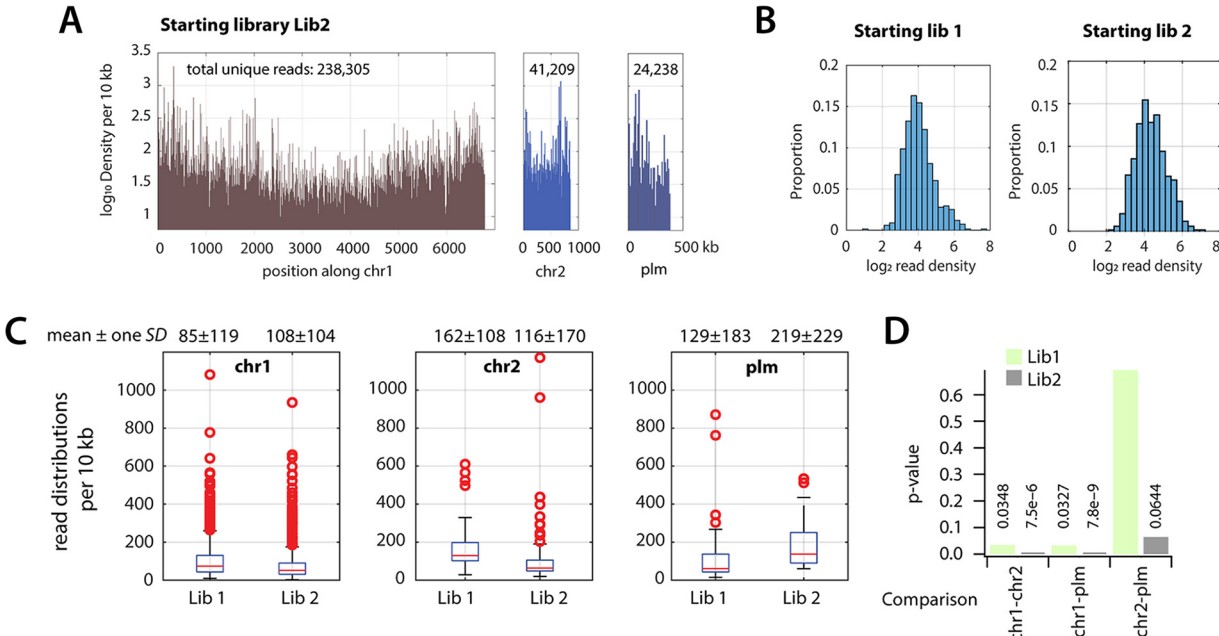

**FIG 1** Gene insertion distributions in *P. veronii* transposon libraries. (A) Log$_{10}$ transposon insertion density per 10 kb in one of the starting libraries plotted as a function of genomic location (chr 1, chromosome 1; chr 2, second replicon; plm, third replicon). Numbers in the upper part of each panel correspond to the total number of unique mapped reads across all samples in the library. (B) Log$_2$ insertion density distributions per 10 kb in chr 1 in both starting libraries. (C) Gene insertion densities per 10 kb across chr 1, and second and third replicons (chr 2 and plm). The values reported at the top of each box correspond to the mean density $\pm$ one SD. (D) P values correspond to *t* test comparisons (two-sided) of all grouped 10-kb densities per sample between Chr1-Chr2, Chr1-Plm, and Chr2-Plm, for both initial libraries.

The unique insertions in the starting libraries mapped to between 77.6% and 78.5% of the predicted *P. veronii* genes (Table S1), with at least a single insertion. For the library samples subsequently transferred in liquid culture (LIQ), between 3 and 5 million cleaned trimmed reads were recovered, of which between 63.6 and 68.4% (Lib1-LIQ) and 24.6 to 47.1% (Lib2-LIQ) were mapped to the *P. veronii* genome (Table 1). Coverage in the second library in toluene-grown liquid culture was lower because of accumulation of a few extreme outliers, as will be explained below. The soil microcosm-grown *P. veronii* libraries (Lib1-SAND and Lib2-SILT) produced approximately 1.7 times less mapped reads (Table 1). Libraries sequenced at later time points on average mapped to fewer genes, indicating fitness selection by the growth conditions but also

**TABLE 1** Sequenced *P. veronii* transposon insertion libraries and sampling times

| Series | Sampling time | Starting library | Description | Generations | Total raw reads[a] | Mapped reads | Mean % genes with insertions |
|--------|---------------|------------------|-------------|-------------|-------------------|--------------|------------------------------|
| Lib1 | T0 | | Starting library 1 | | 2,788,655 | 2,398,050 | 78.5 |
| Lib2 | T0 | | Starting library 2 | | 3,448,875 | 2,151,673 | 77.6 |
| LIQ1 | T1 | Lib1 | 1st cycle[a] | 6[b] | 3,167,777 | 2,666,172 | 63.6 |
| | T2 | | 4th cycle | 24 | 4,481,652 | 3,844,596 | 65.9 |
| | T3 | | 8th cycle | 48 | 5,074,464 | 4,399,844 | 68.4 |
| SAND | T1 | Lib1 | 1st cycle | 9 | 3,172,673 | 2,697,453 | 79.9 |
| | T2 | | 3rd cycle | 15 | 2,561,061 | 2,104,241 | 73.2 |
| | T3 | | 15th cycle | 51 | 1,983,358 | 1,236,390 | 27.0 |
| LIQ2 | T1 | Lib2 | 1st cycle | 7 | 1,286,925 | 1,041,101 | 47.1 |
| | T2 | | 2nd cycle | 14 | 3,869,589 | 3,126,838 | 32.9 |
| | T3 | | 3rd cycle | 21 | 4,852,882 | 3,560,488 | 24.6 |
| SILT | T1 | Lib2 | 1st cycle | 9 | 3,092,741 | 1,991,064 | 71.5 |
| | T2 | | 2nd cycle | 12 | 2,875,037 | 2,106,847 | 63.7 |
| | T3 | | 3rd cycle | 15 | 1,580,122 | 1,249,462 | 47.4 |

[a]Summed amounts from four replicates.
[b]Approximate number of generations of growth of the *P. veronii* metapopulation after that cycle.

in this case because of strong increase of a few highly dominating clones that skewed insertion distributions (see below).

**Essential genes in *P. veronii* starting insertion libraries.** As both starting libraries had been enriched to deprive them of wild-type *P. veronii* and *E. coli* donor cells, we expected that the procedure itself had resulted in selective conditions and loss of certain gene insertions, as shown by Gallagher et al. (42). To identify gene insertions that possibly impaired mutant fitness already in the starting libraries, we scored those genes for which the number of insertions was statistically significantly lower than what would be expected from a library with completely random insertions (considering gene size and the observed 2.4 to 3.0 $\times$ $10^5$ unique insertions per genome in the starting libraries, summed per 10-kb regions). At a ratio <0.05 of observed to expected insertions and a false-discovery rate (FDR) <0.005, the initial libraries overlapped in 778 genes, of which the majority mapped in the chromosome 1 replicon (Fig. 2A, Table S2). A total of 289 of those overlapped with the essential gene set ($n = 565$, 440 orthologous to *P. veronii*) identified from *Acinetobacter baylyi* ADP1 for growth on minimal medium (42). Several conspicuous gene regions into which marker insertions were expected to cause fitness loss, such as cell division (*fts*) and ribosomal protein synthesis (*rpl*, *rps*, Fig. 2B), were indeed strongly depleted for insertion densities in both starting libraries. The list of common essential genes at this threshold encompassed all attributable categories in the Cluster of Orthologous Groups (COG) classification (Fig. 2C). However, COG classes C (energy production), J (translation), and M (cell wall and membrane biogenesis), and to a lesser extent classes D, H, and L were statistically significantly overrepresented in comparison to COG attribution from randomly drawn gene sets of the same size as the list of essentials (FDR < 0.05, $n = 10$; Fig. 2C). In contrast, notably classes COG-N (cell motility), -P (inorganic ion transport and metabolism), -R (general function prediction), -T (signal transduction mechanisms), -V (defense mechanisms), and -X (mobilome) were underrepresented compared to random picking from the genome, suggesting fewer general negative fitness effects of insertions in these categories (Fig. 2C).

Seen at the level of metabolic pathways, particularly the biosynthesis of aminoacyl-tRNAs, fatty acids, lipopolysaccharides, peptidoglycan, pantothenate, and coenzyme-A were affected more than expected by chance, as well as the metabolism of folate, nitrogen compounds, vitamin B6, glycine, serine and threonine, and of the tricarboxylic acid cycle (Fig. 2D). Essential gene attribution to biosynthetic and metabolic pathways of other amino acids, nucleotides and carbohydrates was not significantly different from random models (COG-E, -F and -G; Fig. 2C). This suggests such mutants to survive to some extent in the "metapopulation" of *P. veronii* mutants (which was grown as a pool), perhaps because of metabolite sharing, or due to pathway redundancies.

**Gene insertion fitness effects as a function of population dynamics and growth conditions.** In order to identify genes with conditional fitness effects, we followed and contrasted the growth of the *P. veronii* mutant libraries under two different culturing conditions (nonsterile soil and sterile liquid microcosms). We added (gaseous) toluene as a specific carbon substrate for *P. veronii* to favor its growth in soils and imposed multiple culturing cycles to obtain up to an estimated ~50 generations of metapopulation growth (Table 1). Two types of soil were selected, to observe more general as well as specific effects, with the Lib1 metapopulation propagated in sandy soils (Lib1-SAND) and Lib2 in silt (Lib2-SILT). Our hypothesis here was that the specific culturing regimes would create selective effects, leading to a gradual change in the relative proportions of individual *P. veronii* mutants in the starting libraries. This would be reflected in changes in relative proportions of per-gene insertion read densities in the sequenced libraries at each sampling time point. We expected further that growth in soil would be in some ways like that in liquid because of the added toluene and the mineral salts, but that specific soil selection effects might be distinguished by intersecting with those observed in the corresponding liquid controls.

The Lib1 *meta*-mutant populations repeatedly grew to similar densities in sand microcosms during multiple dilutions and across the four replicates (Fig. 3A). Cells were recovered and insertion positions sequenced after an estimated 6, 24, and 48 generations in

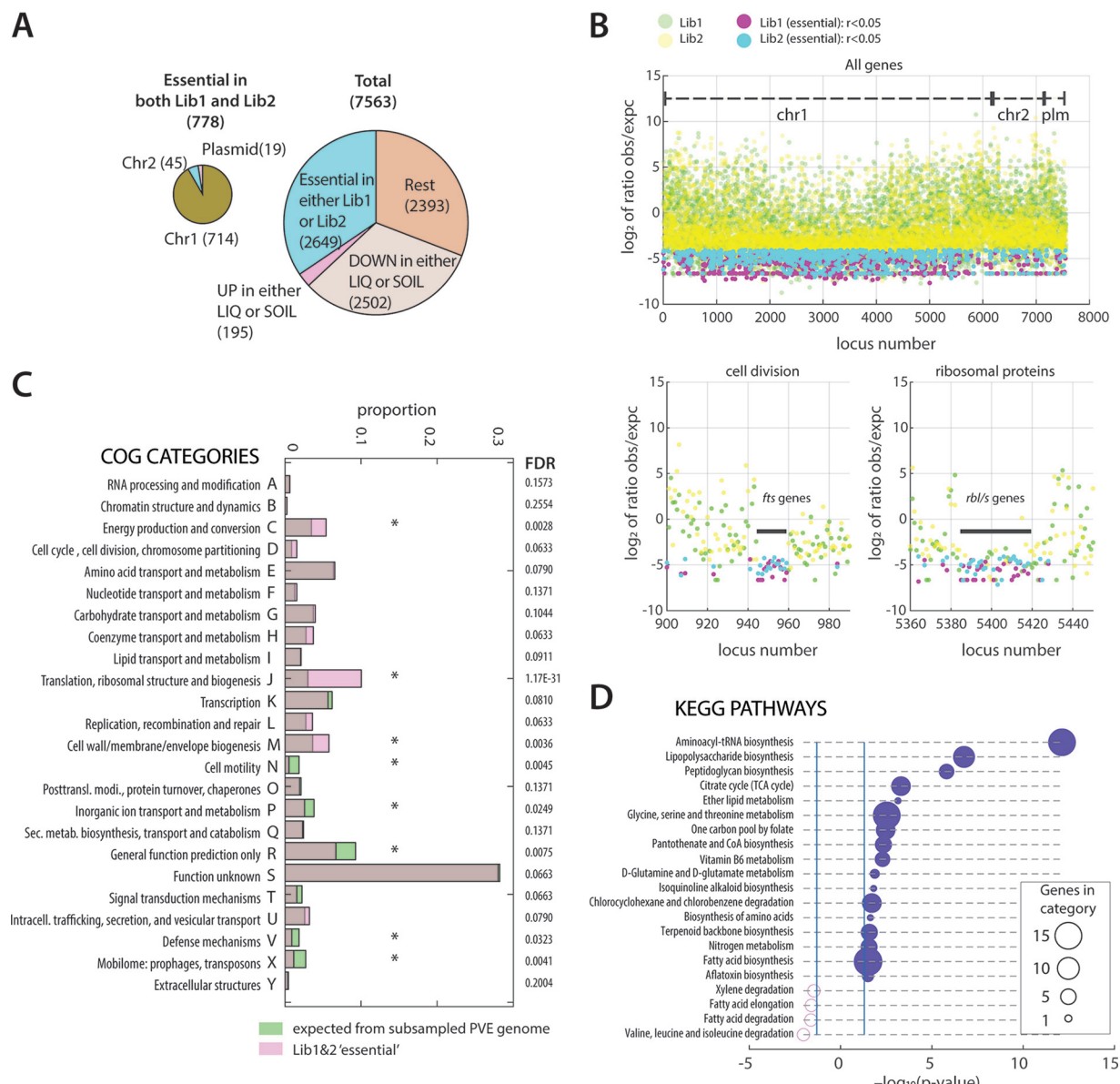

**FIG 2** Global analysis of essential genes in *P. veronii*. (A) General overview in numbers of affected genes in the different used categories or replicons. Total: total of identified genes in *P. veronii*. (B) Scatterplot of the log$_2$ ratio of mean observed/expected reads per 10 kb in the starting libraries, plotted according to gene number along the genome (chr1, chr2, and plm). Each dot represents a gene; Lib1, green; Lib2, yellow. Thresholds: Genes with observed/expected ratios ($r$) in both starting libraries <0.05 for Lib1 (magenta) and Lib2 (cyan). Zoomed-in panels illustrate absence of insertions in gene regions of the mostly essential cell division genes (*fts*) and ribosomal protein coding genes (*rpl-rps*), respectively. (C) Cluster of orthologous group (COG) attribution among the essential genes overlapping in Lib1 and 2 from panel A (transparent magenta), compared to $n = 10$ randomly subsampled gene lists from the full genome with the same size (transparent green). Asterisks indicate significant enrichment or depletion at an FDR < 0.05. (D) Enrichment (filled circles) and depletion (open) of KEGG-pathway attributions from the Lib1 and 2 essential gene list of *P. veronii*. Circle sizes show the number of genes in the corresponding KEGG pathway category. *P* values calculated from the observed attribution variation compared to that from $n = 100$ randomly sampled gene lists of the same size from the *P. veronii* genome (778 genes). Cutoff for display at $P < 0.05$.

liquid (Lib1-LIQ), or 9, 15, and 51 generations in sand (Lib1-SAND; Table 1). In contrast, Lib2 mutant populations cultured and passaged in silt microcosms maintained similar population densities only for the first three passages, after which they decreased almost to extinction (Fig. 3B). Consequently, we restricted our sampling to the first three cycles with an estimated 9, 12, and 15 generations of growth (Lib2-SILT), and 7, 14, and 21 generations in liquid (Lib2-LIQ; Table 1). The reason for this difference and the almost complete extinction of *P. veronii* Lib2 over time in silt is not clear. Silt has a 5-fold higher cell count than the used sand ($10^9$ versus $2 \times 10^8$ per gram) (54), and a higher taxa diversity (Fig. 3C, Table S3). Bacterial taxa at the first sampling point in the soil microcosms (9 generations)

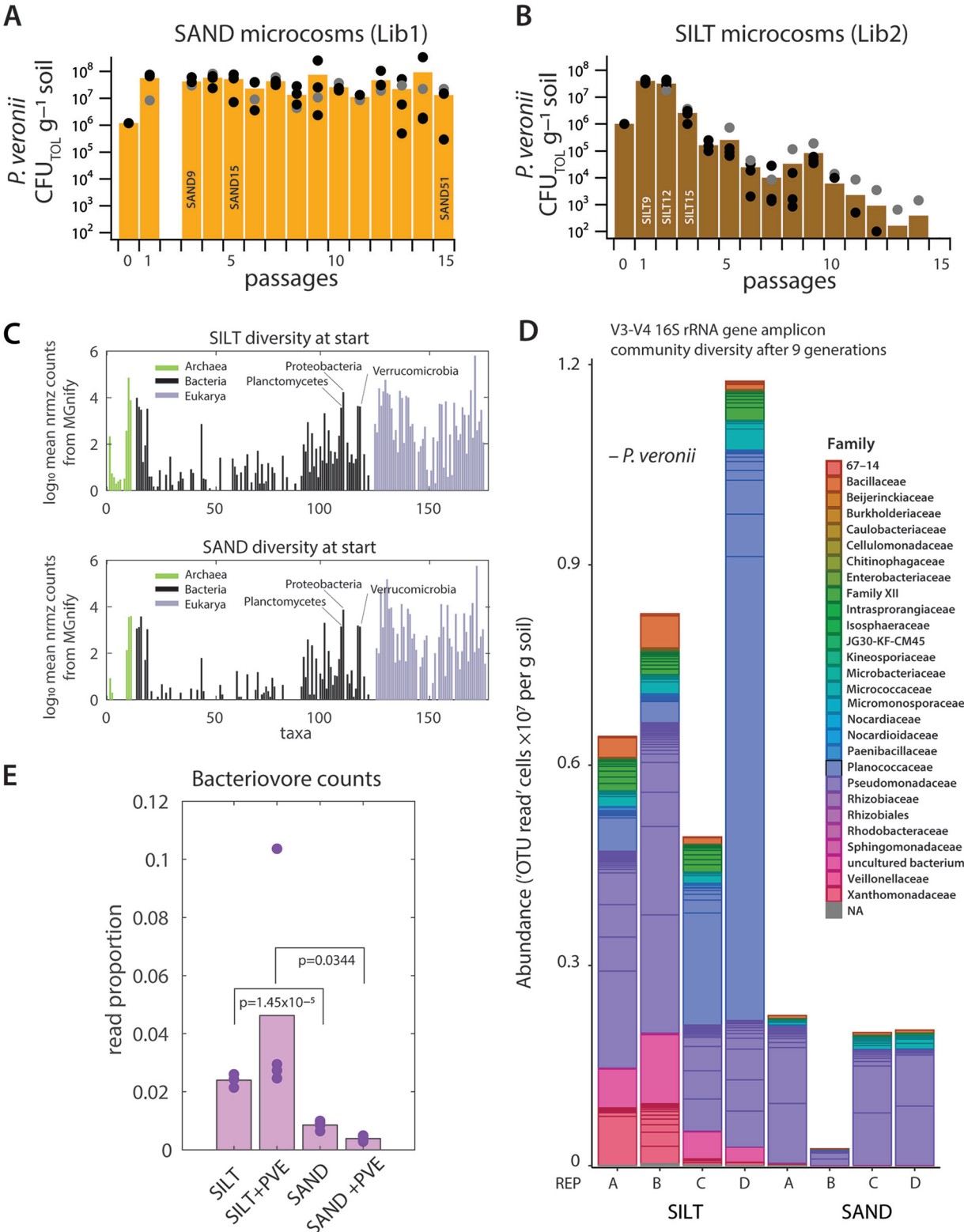

**FIG 3** *P. veronii* transposon mutant library growth in nonsterile soil microcosms. (A and B) *P. veronii* metapopulation sizes after each passage in sand (A) or silt (B) microcosms under toluene vapor. Soil growth cycles consisted of a 1:10 wt/wt dilution into fresh soil. CFU: CFU on plates with toluene vapor. SAND9, etc.: estimated number of generations of growth of the *P. veronii* metapopulation and the deployed sampling points for library sequencing. (C) Taxa diversity (Bacteria, Archaea, and Eukarya) from metatranscriptomic sequenced sand and silt starting material, estimated by the MGnify pipeline. Bars represent the log₁₀ mean summed phyla counts across four replicates, normalized to the total number of taxa-attributed counts. Illustrative names are indicated (full list in Table S3). (D) Proportional bacterial taxa abundances in Lib2-inoculated silt and Lib1-inoculated sand microcosms deduced from V3-V4 16S rRNA gene amplicon sequencing (*P. veronii* reads removed).

are still more diverse in the Lib2-inoculated silt than Lib1 in sand (Fig. 3D). Both become dominated by *Pseudomonadaceae* (Fig. 3D), possibly as a result of direct or cross feeding with *P. veronii* on the added toluene. Previously, we noted some 10% non-*P. veronii* CFU appearing on toluene-selective plates among cells extracted from *P. veronii* wild type-inoculated silt but not sand microcosms (54), suggesting direct substrate competition for toluene in silt. Finally, we also measured between 2.8 and 12 times higher read counts of bacterivorous soil eukaryotes in silt than in sand samples at start (Fig. 3E, Table S4). This may be a sign of increased predation from native protists on the inoculated *P. veronii* in silt as opposed to sand, which contributed to their decline.

**General conditional fitness effects.** Replicate Tn-seq samples grouped close together in multidimensional scaling and further clustered by treatment and time, and samples taken after more generations of mutant library growth, tended to drift "further" from the starting libraries (Fig. 4A). Replicate variability on average resulted in a 10% variation of the mean insertion density per gene, with notable exceptions at the highest generations in sand (Lib1-SAND, 51 generations) and in liquid suspension (35 to 100% variation; Fig. S1). All four treatments resulted in the appearance of a few clones that dominated over 50% of all reads at the later time points (Fig. 4B, Table S5), suggesting they provide large fitness advantages under the tested conditions. Some of those, e.g., PVE_r1g4308 (*fleQ*) or PVE_r2g382 (*ttgV*, toluene efflux pump regulator), appeared consistently across both libraries and in both liquid and soil growth conditions (Fig. 4B). Other gene insertions (e.g., PVE_p353, RepB-family plasmid replication initiator protein; and PVE_p354, hypothetical protein) became enriched only in one of the libraries. Further, it is interesting to note that 7 out of the 12 extreme outliers were comprised of genes found on replicons chr2 and the plasmid in *P. veronii*. For further analysis, these outliers were removed from all the libraries and all conditions to avoid comparing skewed distributions. Notably, the effect on soil fitness of the *fleQ* gene was studied separately on a clean gene deletion mutant produced by recombination marker exchange (see below).

To better discern consistent conditional fitness effects of gene insertions in the two libraries, we took advantage of the three-time/generation sampling points. After outlier removal, the read count libraries were normalized to insertion densities per gene ($n = 4$ replicates; using the PseudoReference normalization procedure as specified in the Materials and Methods section and in [57]) and then subsampled according to the observed read distributions per sample to obtain the same read sums. This was used to average gene insertion counts combined for all three sampling time points and proportionate to the corresponding starting sample, from which the adjusted $P$ values of per-gene differences were calculated. Insertion counts over time were subsequently scored as *down* if their mean ratio compared to start was less than 2-fold and with FDR < 0.05; and as *up* with a ratio to T0 > 2 and FDR < 0.05. From both categories, we excluded those genes categorized as essential (established above).

Taken together, 2,502 genes classified to *down* and 192 to *up* in either of the liquid or soil microcosm samples, whereas insertion counts in 2,393 genes remained without statistically significant change (Fig. 2A). On average, more fitness effects were detected in Lib1 than Lib2 samples (Fig. 5A). This might be due to the stronger primary selection during Lib2 preparation (minimal medium [MM] with succinate) than Lib1 (lysogeny broth [LB] medium), because of which many insertions with fitness effects on minimal medium growth (in Lib2) had already been counterselected before being transferred into the microcosms. Furthermore, as we expected, a large proportion of *down* insertions overlapped between the soil and liquid incubated library samples (Fig. 5B). Growth and propagation in liquid overall represented a stronger fitness selection, given the higher number of gene insertion depletions than in the corresponding soil microcosm incubations (both for sand and silt;

**FIG 3** Legend (Continued)

Operational taxonomic unit (OTU) attributions are displayed as abundances (OTU reads per gram soil) by normalizing to the *P. veronii* reads and their mean measured culturable density in the samples (not corrected for potential differences in 16S rRNA gene copy numbers per genome). (E) Summed read proportions of eukaryote bacterivores in sand or silt, in presence or absence of inoculated *P. veronii*, deduced from metatranscriptomic data attributed to taxa by MGnify (Table S4).

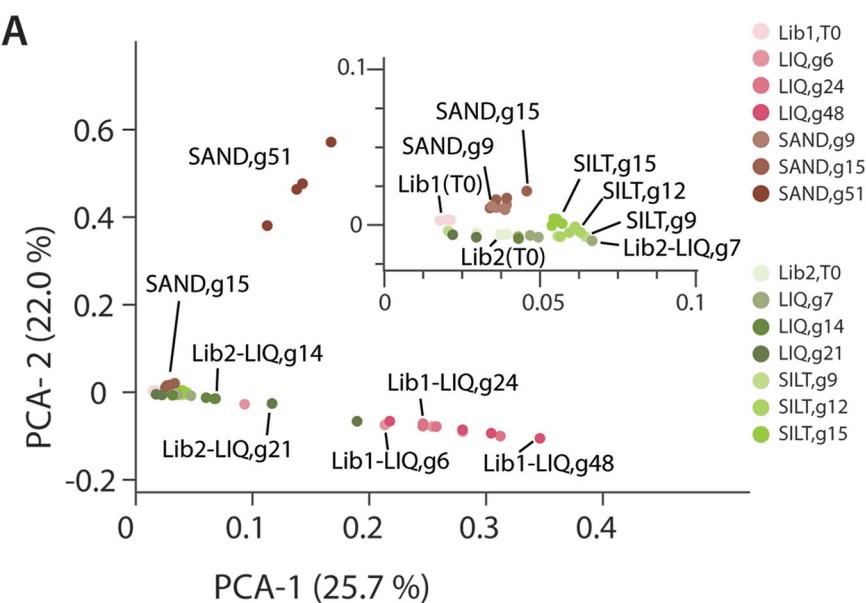

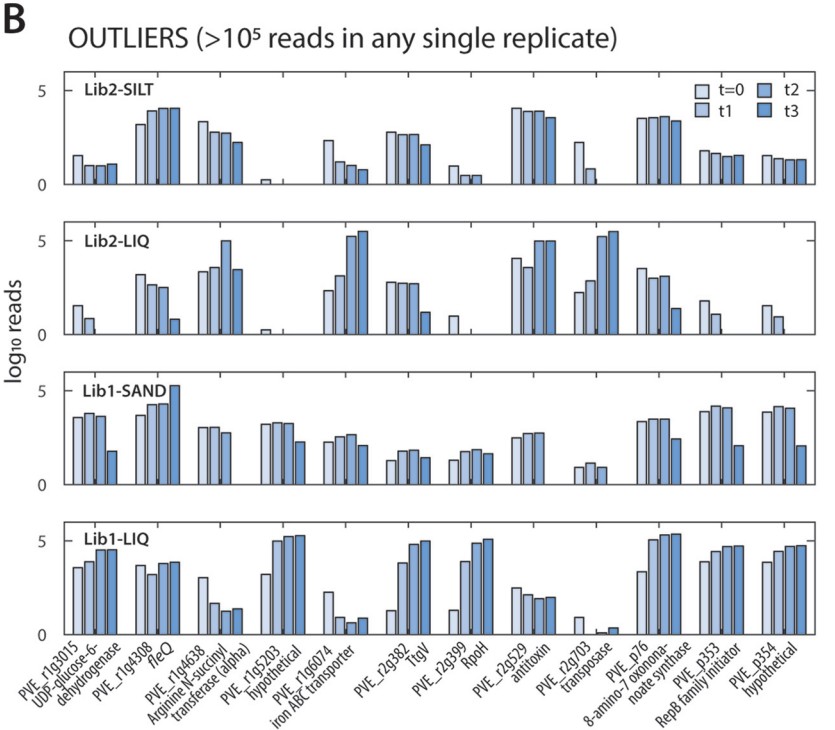

**FIG 4** Global library sample variation and appearance of extreme outliers. (A) Principal-component analysis of the gene insertion counts in the different libraries, conditions, and replicates. (B) Identified outliers of the 0.01% gene insertions making up more than 100,000 reads individually. Bars display mean $\log_{10}$ insertion counts from four replicates at the indicated locus over the four sampling points (increasing blue hues).

Fig. 5B). This suggests that the soil environments (despite being replenished with the same minimal medium components and carbon substrate as the liquid) have other nutrients available that support proliferation of *P. veronii* insertion mutants or provide spatial separation from competitive effects.

**Distinguishing soil microcosm and liquid growth conditional fitness effects.** To tease out fitness effects for proliferation in either liquid or soil, we focused on the respective defined common and exclusive gene sets (Fig. 5A and B). Because of limited direct

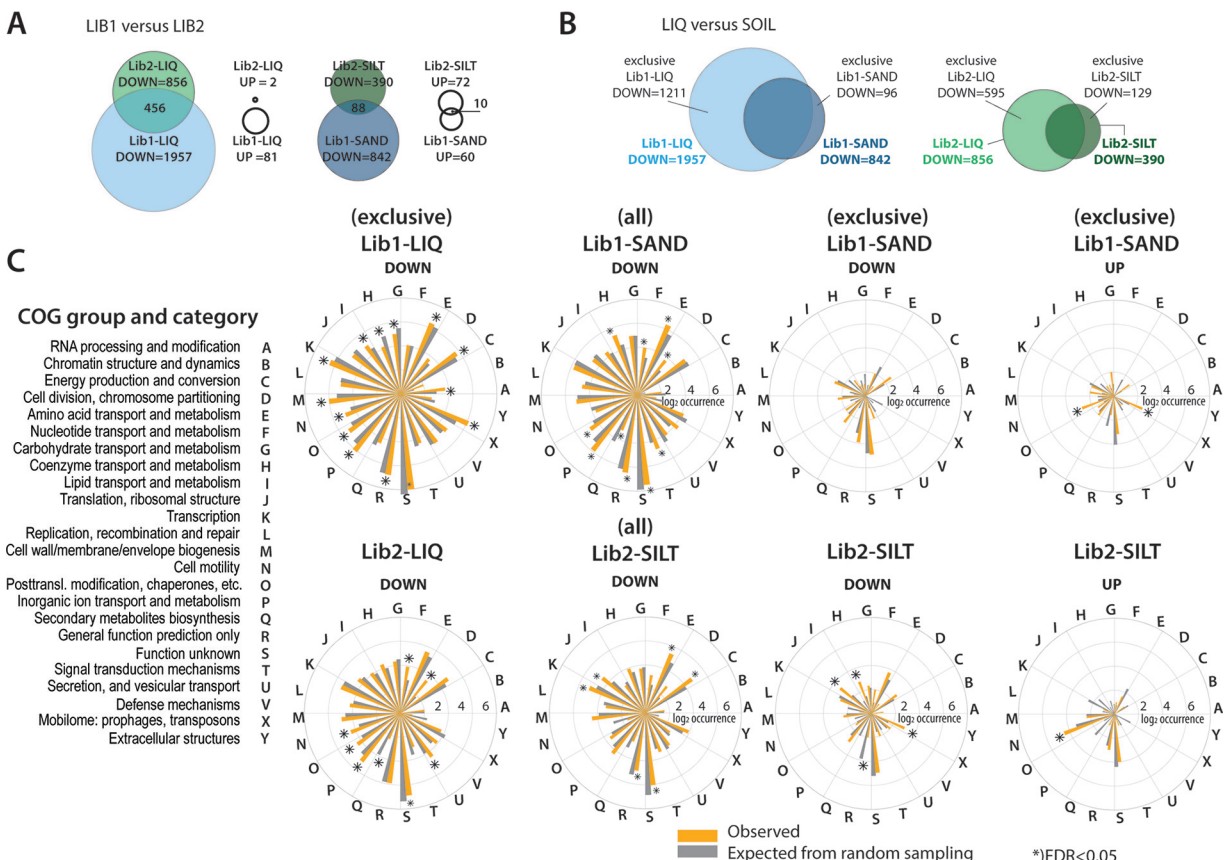

**FIG 5** Common and condition-specific fitness-affected gene groups of *P. veronii* during growth with toluene. (A) Comparison of the number of *up* (enriched) and *down* (depleted) gene insertions for the two libraries and their overlap. (B) As in A, but between liquid and soil conditions within each library. "Exclusive" refers to genes unique for either liquid or soil conditions. (C) COG class attributions for each of the libraries, the specific growth conditions (LIQ, liquid growth; SILT or SAND, soil microcosm growth) and the direction of fitness attribution (UP or DOWN; lists and names as defined in A and B). Bars show the mean observed class attribution (as log₂ occurrence, ochre) versus the mean expected attribution from 10 random genome samples of genes with the same sampling size (gray). Asterisks indicate significant enrichment or depletion (at FDR < 0.05) compared to randomly drawn.

overlap in gene lists between Lib1 and Lib2 samples, we compared COG class attributions, assuming similar global effects would become selected even when individual gene insertions are not the same. COG attributions were next placed in perspective with attributions from repeated randomly sampled *P. veronii* gene lists of the same size (Fig. 5C). Both distributions of COG class attributions of *down* insertions were very similar between the exclusive liquid (Lib1-LIQ down and Lib2-LIQ down; Fig. 5C) and the (total) soil *down* gene lists (Lib1-SAND, Lib2-SILT). They were also more or less congruent with the expected randomly sampled COG distribution of the *P. veronii* genome as a whole (gray bars in Fig. 5C). Notable differences concerned, for example, COG classes G, H, and I, which were significantly more frequently depleted in the Lib1 than in Lib2 samples (possibly for reasons of the different library production; LB for Lib1, minimal medium for Lib2) (Fig. 5C). No particular COG class was enriched in the exclusive *down* gene list of Lib1-SAND; however, COG-H (coenzyme transport and metabolism), COG-J (translation and ribosome structure), and COG-X (mobilome) were enriched in the exclusive Lib2-SILT *down* gene list (Fig. 5C). Interestingly, both soil-passaged *P. veronii* mutant libraries became significantly enriched in gene insertions in COG group N (cellular motility functions; Lib1-SAND *up* and Lib2-SILT *up*) (Fig. 5C). No overlap, nor COG category enrichment or depletion was detected among the *up* lists of the liquid samples (Fig. 5A).

Because the main COG-classification does not have a high resolution, we further compared the exclusive and overlapping *down* lists in soils and liquid at the level of Kyoto Encyclopedia of Genes and Genomes (KEGG)-attributable metabolic pathways (58). As expected from the COG analysis of Fig. 5C, depleted gene insertions both under liquid and

soil growth encompassed a wide range of metabolic pathways, such as those involved in amino acid, nucleotide, or cofactor synthesis, energy generation, and carbohydrate, aromatic compound, and lipid metabolism (Fig. 6A). The covered pathways were indeed largely similar between the two libraries, even though the overlap of individual genes was limited, as mentioned. This indicates similar selective constraints presented by the same growth conditions on the two independently generated libraries. As the COG-categorization had illustrated, growth in liquid exacerbated the extent of mutant depletion in most of the pathways, compared to soil microcosm incubations. This is a strong indication that many conditional fitness mutants for growth in minimal medium can still survive and proliferate in soils.

In contrast to our initial expectations, very few pathways were specifically highlighted among the exclusive soil *down* categories (Fig. 6A, magenta lines). Although genes exclusively being depleted in silt (Lib2-SILT *down*; Fig. 6A) seemed to lead to more pathway "defects" than in sand (Lib1-SAND *down*; Fig. 6A,), visual inspection suggested several of those genes (Lib2-SILT) to be already present among the liquid *down* category (Lib1-LIQ). This was further quantified by calculating the potential pathway enrichment among the *down* gene lists against a random model (i.e., subsampling genes from the *P. veronii* genome with the same size as the comparison group and attributing them to KEGG pathways; Fig. 6B). Distinct differences could be observed between soils and liquid, for example in KEGG pathways *Alanine, aspartate and glutamate metabolism*, *Glycerophospholipid metabolism*, *Butanoate* and *D-alanine metabolism* (pathways numbered 1, 7, 9, and 12; Fig. 6B), whereas others were indeed cross-wise overlapping between Lib2-SILT and Lib1-LIQ, as noted by visual inspection, or vice versa (e.g., KEGG pathways *Cysteine and methionine* [8], *Nicotinate and nicotinamide* [15] and *Pyruvate metabolism* [19]; Fig. 6B).

Since all experimental conditions included the presence of toluene as growth substrate, we expected to see fitness differences for insertions in genes for the known aromatic metabolic pathways of *P. veronii*. Insertions in the *ipbAa-Ad-B-C* genes encoding the first step of toluene degradation were disproportionally low, even in the starting libraries, but their densities diminished further to complete absence under all conditions (Fig. 6C). Insertions in the *dmp* and *ibpEGFD* genes for the *meta*-cleavage branch were less drastically affected than *ipbAa-Ad-B-C*, possibly because they are functionally redundant; but they also decreased at later time points (Fig. 6C). In contrast, insertions in the *nah* gene-encoded *meta*-cleavage pathway were mostly without measurable fitness cost in soils (except for the last sand time point), but were more affected in liquid growth conditions (Fig. 6C). This suggests they are either not expressed or important for the metabolism of other aromatic substrates for which our microcosms did not select.

**Soil conditions select for defects in motility.** The enrichment of gene insertions in cell motility functions in soil-grown mutant libraries suggested by the COG categorization (Lib1-SAND *up* and Lib2-SILT *up*; Fig. 6C) was largely due to insertions in genes for flagella and flagellar motor biosynthesis, which collectively increased between 3 (after 15 generations) and 30-fold (in sand, after 51 generations; Fig. 7A). Although in majority confined to a single large consecutive gene region on chromosome 1 (PVE_r1g4273-PVE_r1g4604), the enrichment under soil conditions was also visible in other flagellar annotated genes (e.g., PVE_r1g1825, PVE_r1g3039; Fig. 7B). Notable exceptions to the enrichment were genes for flagellin itself (PVE_r1g3212 and _r1g4313; Fig. 7B), suggesting that it is more the biosynthesis and functioning of the flagellar motor that is counterselected, rather than the presence of the flagella. This enrichment of flagellar biosynthesis and motor gene insertions contrasted to other well-defined biological machines, such as the type VI secretion systems (Fig. 7C) or genes involved in chemotaxis, type III secretion, conjugation, or denitrification (Fig. S2). The growth advantage of flagellar-mutants was confirmed by an independently reconstructed deletion of the *fleQ*-gene (PVE_r1g4308), which was indeed nonmotile and showed a statistically significantly faster growth than wild-type *P. veronii* both in nonsterile

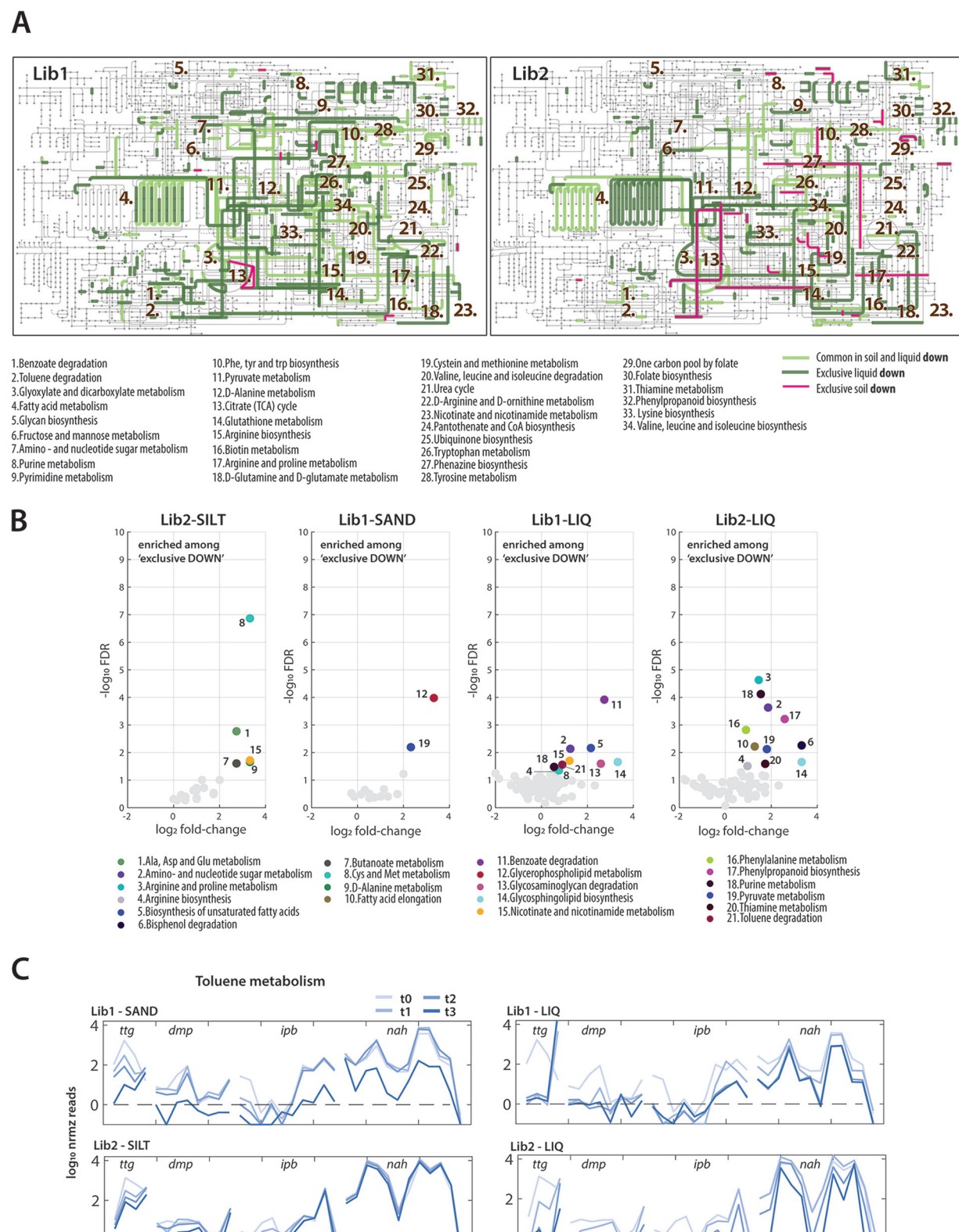

**FIG 6** Metabolic pathway attribution of *P. veronii* depleted gene insertions during liquid or soil growth. (A) KEGG pathway attribution of the *P. veronii* genes commonly depleted in liquid and soil (light green), exclusive liquid (dark green), or exclusive soil (magenta). Numbers correspond to pathway names as in the bottom legend. (B) Significance of KEGG pathway enrichment compared to randomly sampled gene

silt (with toluene as added C-source), as well as in regular liquid suspended growth with succinate (Fig. 7D).

**Specific fitness decrease in soil incubations.** From the exclusive *down* gene lists of Lib1 in sand and Lib2 in silt, we manually curated those groups of gene insertions, which are common to both (28 genes) or specific for either soil (Lib1-SAND: 36 and Lib2-SILT: 41; Table S6), while not being listed under any of the liquid growth conditions. Their gradual and soil-specific loss over increasing generation times of library incubation is apparent, in contrast to the increase of the flagellar genes and the more neutral behavior of insertions in the type VI secretion system (Fig. 8A and B). The common depleted gene insertions cover several transport systems (e.g., for nitrate, sulfate, galactonate, carbonate; Table S6), which may contribute to *P. veronii* fitness while growing in soils. More specifically, growth in sand led to depletion of insertions in five regulatory genes and two cell wall functions (a cell wall hydrolase, PVE_r1g0562; and a penicillin-binding protein 1C, PVE_r1g0671; Table S6). Growth in silt, on the other hand, led to depletion of insertions in further transport systems (e.g., zinc, iron, organic acids, and amino acids), cell wall and lipopolysaccharide modifications, and different transposases (Table S6).

## DISCUSSION

Bioaugmentation relies on the capacity of exogenous strains with xenometabolic potential to adapt, grow, and survive after inoculation in a polluted environment and within a resident microbiome (1, 2). However, it is still rather unclear which inherent characteristics of strains contribute to their intended growth and survival, particularly in soils (4). The major aim of the underlying study was to identify genes in the candidate strain for monoaromatic compound bioaugmentation agent *P. veronii* 1YdBTEX2 that are crucial for fitness maintenance in soil under pollution stress. By generating two independent high-density genome-wide random insertion libraries, each contrasted in growth cycles in two different soils (artificially contaminated with toluene) and compared to their toluene liquid growth control, we broadly define the groups of genes whose fitness is affected under both conditions, as well as insertions that influenced fitness more specifically in the (toluene-contaminated) soils. Common fitness-affected categories in liquid and soil included a wide range of functions, such as inorganic ion transport, fatty acid metabolism, amino acid biosynthesis, and nucleotide, thiamine, and other cofactor metabolism. The extent of affected pathways in libraries grown in soil, however, was only half of that in liquid. This indicates that, in general, proliferation in soil is more supportive than in minimal medium liquid, probably because of availability of additional carbon substrates and nutrients. Proliferation in soils, surprisingly, led to reproducible strong fitness gain of *P. veronii* mutants with insertions in genes for (the) flagellar motor complex(es). The number of genes with insertions exclusively affecting soil growth and survival was relatively sparse, indicating that *P. veronii* rather adapts to the presence of the carbon and nutrient substrates, irrespective of the environment it is in, provided the absence of too severe obvious water or toxic stresses (33). This supports previous conclusions drawn from *P. veronii* exponential and stationary-phase transcriptomics measurements in different nonsterile soils (54), that had also suggested streamlined and consistent toluene metabolism independently of the growth environment. This environment-independent physiological streamlining of *P. veronii* contrasts to the proposed soil-specific growth program observed in the dibenzofuran-degrader *Sphingomonas wittichii* (48), which may thus point to different evolved ecological strategies.

Tn-seq is extremely powerful to identify conditionally dependent fitness effects and has been widely applied to understand gene essentiality under different growth condi-

**FIG 6** Legend (Continued)

lists of the same size as the observation group (FDR < 0.05 and 2-fold increase). Numbers and colors correspond to KEGG names listed below. (C) Mean $\log_{10}$-normalized insertion counts for genes implicated in toluene metabolism in *P. veronii* (i.e., *ttg*, *dmp*, *ipb*, and *nah*), averaged from the four individual library replicates, sampled at start and at three different time points (increasing hues of blue), for the two libraries and soil or liquid growth conditions.

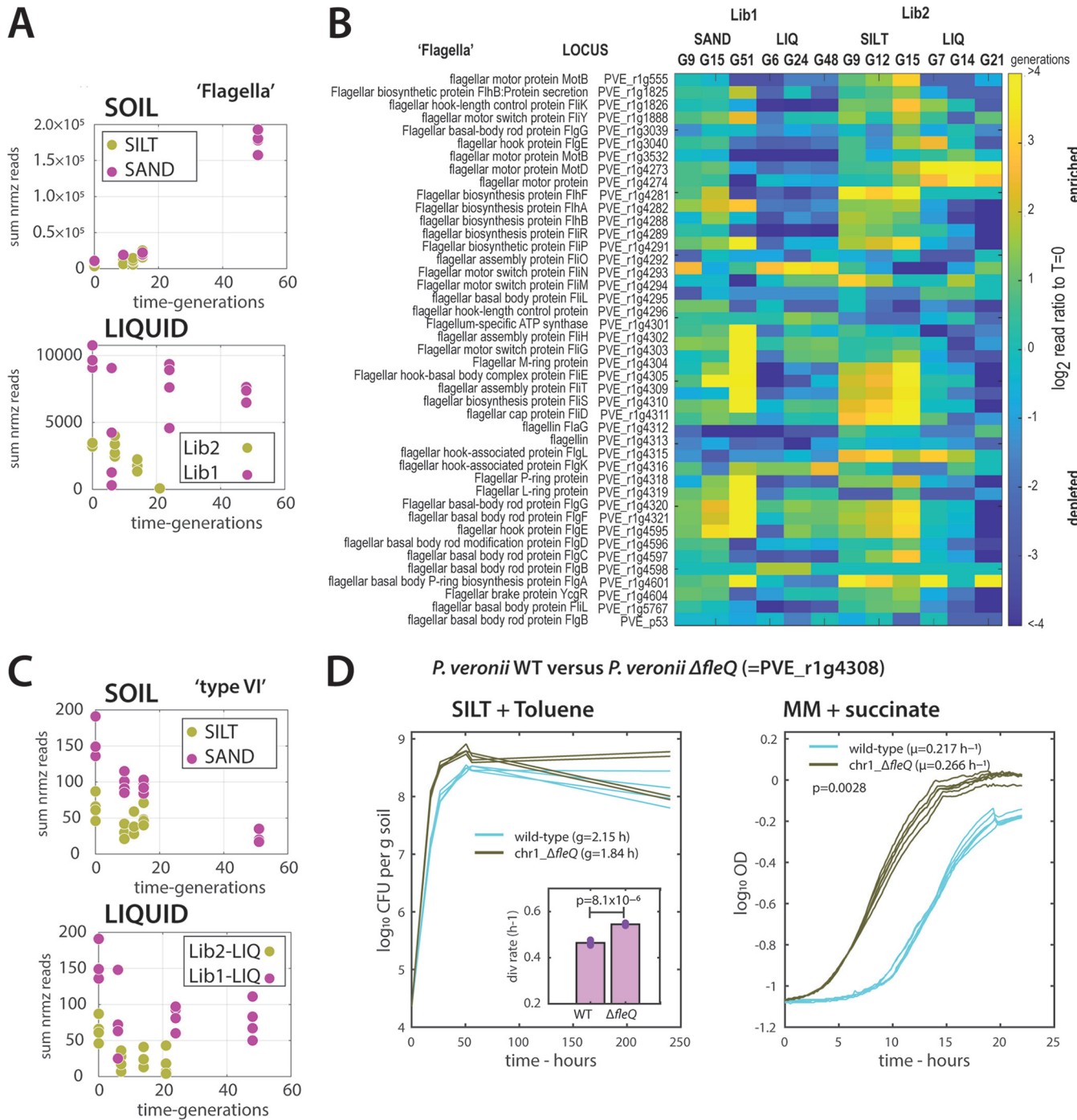

**FIG 7** Fitness increase for *P. veronii* flagellar mutants during growth in soil with toluene. (A) Summed normalized read counts of insertions into genes annotated to flagella, for the *P. veronii* libraries (Lib1 and Lib2, corresponding colors in both panels) incubated in either of the two soils or in liquid suspended cultures. (B) Log₂ ratios of mean gene insertion counts (4 replicates) after each time sample for each of the libraries and conditions compared to its respective start, depicted as a heatmap with scale as shown on the right. Note the fitness increase for insertions into genes for flagellar motor, basal body, or hook, but not the flagellin. (C) Summed normalized read abundances for gene insertions for type VI secretion systems, as an example for a biological machine without measurable fitness difference in soils or liquid. (D) Growth rate increase of a reconstructed *fleQ* flagellar regulator deletion mutant (PVE_r1g4308) in nonsterile silt microcosms (with toluene as specific added C source) and in liquid suspended culture with succinate. Lines show four (microcosms) or six (liquid) replicate population size measurements. Inset shows the mean calculated cell division rate and the two-sided paired *t* test-derived *P* value.

tions in pathogenic bacteria (46, 47, 59–61). It has so far been less frequently deployed for environmental bacteria with bioaugmentation potential (43–45, 48). The technique also poses several challenges and pitfalls both in experimental designs as well as in the generation of the insertions that complicate the quantitative analysis of inserted marker

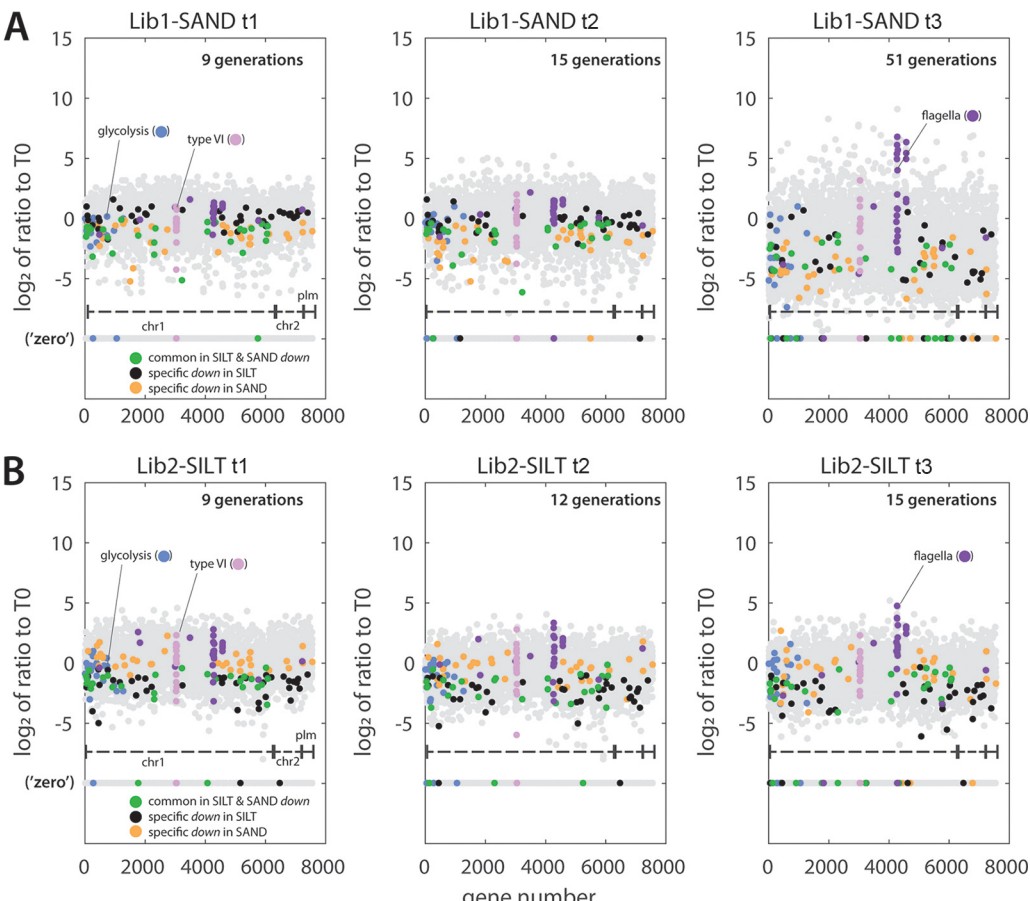

**FIG 8** Time evolution of fitness change in genes exclusively attributed to soil conditions. (A and B) Log$_2$ ratio plots of the mean insertions (four replicates) at the indicated sampling time to that at incubation start (T0). Each gray dot is a value from a single gene, organized as function of gene number along the *P. veronii* genome (chr1, chr2, and plm). Overlaid are categories identified in the analysis above, e.g., flagellar genes, type VI secretion system genes, genes commonly depleted in both soils, and genes exclusively depleted in either silt or sand microcosms (according to the color scale and for either the Lib1 samples in A or Lib2 samples in B).

frequencies in analyzed libraries. Depending on the host strain and the characteristics of the (hyperactive) transposon, target insertion positions may not be completely unbiased. The relaxed Tn5 transposon used here is relatively sequence unspecific, which the sparse overlap (~10%) found among the Lib1 and Lib2 unique insertion positions (500,694 in total) attests. Furthermore, marker insertions can have polar effects on downstream-located genes (62), or may affect the gene product's functionality in different ways depending on their insertion position (41). For these reasons and to reduce the risk of false interpretations, we focused here not so much on single gene effects but rather on functionalities of groups of genes (e.g., metabolic pathways or multicomponent biological functions), and their statistical enrichment or depletion compared to randomized models. Indeed, we capture a range of consistent effects even though individual gene insertions have limited overlap between both insertion libraries.

Finally, complete absence of any selection during the production of the Tn-seq libraries is almost impossible to achieve; therefore, even the first-time sample of insertion positions and abundances is already biased for the absence of highly essential genes (42). Then, depending on the study aims, the libraries are proliferated under the test conditions and their relative proportion of gene insertions is compared to the start. However, as we show here by two different production regimes (LB for Lib1 and MM with succinate for Lib2), the need to culture the initial cells to select for the insertion mutants is influencing the initial relative insertion proportions. Therefore, it is important not just to compare

to an initial library composition, but to the same library incubated under control (here, liquid growth with toluene) versus test conditions (here, soil microcosms).

Despite variation in unique insertion positions and gene insertion frequencies among the two independent transposon libraries, the assessment of initial gene essentiality in *P. veronii* corresponded reasonably well with a recent list of true essential genes identified in *A. baylyi* ADP1 by transposon transformation without selection bias (42). Unfortunately, transformation efficiency in *P. veronii* is relatively poor and we had to rely on conjugation of the transposon plasmid from an *E. coli* donor. Gene essentiality scoring was then based on comparing time-zero observed to expected *in silico* random insertion frequencies with the same density in the *P. veronii* genome, rather than scoring only insertion absence, which proved not to be a good strategy. The list of common insertion-depleted *P. veronii* genes at start between both libraries largely overlapped with associated assumed essential functions, such as cell division, energy production, protein translation, cell wall and membrane biosynthesis, or cofactor biosynthesis, which gave confidence to the fidelity of our analysis. On the other hand, functional gene groups associated with cell motility, inorganic ion transport, defense mechanisms, signal transduction, or the mobilome, were underrepresented at start. This also makes sense, given that such functions are frequently considered auxiliary or form part of the pan-genome variability.

Our main goal was the potential identification of genes important for *P. veronii* fitness specifically in soils under pollutant stress. For this, we applied a conservative data analysis method that took time trends into consideration, rather than a single incubation time point comparison to the starting libraries. Furthermore, we produced two independent libraries that were screened in two different soils artificially contaminated with toluene, which were each compared to a minimal medium liquid suspension control. This was necessary to avoid detecting the influence of gene insertions, which are not exclusive for proliferation in soil, but generally important for growth in minimal medium. We acknowledge that addition of toluene may have biased detection of fitness factors implicated in soil survival, *per se*, but this was necessary to allow sufficient *P. veronii* growth in the soil microcosms, without which we would have been unable to attain the number of generations necessary to detect depletion of gene insertions and deduce fitness effects. The reason to choose two different nonsterile soils was that we expected to see biological differences and hoped to be able to potentially identify factors, which would relate to *P. veronii*'s competitiveness.

Indeed, growth cycling in one of the soils (i.e., silt) led to slow extinction of *P. veronii* after an estimated 15 generations. This indicated much poorer proliferation of the *P. veronii* mutant metapopulation in silt than sand, which likely has its underlying causes in biological differences of the resident material. For example, resident aromatic compound-degrading bacteria may have competed with *P. veronii* for the primary added substrate toluene and may have become enriched over multiple soil transfer cycles. Also, we cannot exclude that the *P. veronii* toluene pathway genes (e.g., *ipb*, *dmp*; all residing on the chr2-replicon), were transferred more efficiently by conjugation to the higher density of resident bacteria in the silt than sand. By itself, the higher number of resident bacteria in silt may also have contributed to stronger competition than in sand. Metabolites are excreted during growth of *P. veronii* on toluene (Hadadi et al. [55] lists such metabolites and concentrations), which may have been utilized by resident bacteria in silt and reduced the *P. veronii* yields. Finally, the grazing pressure by protozoa was likely higher in silt than sand and may have contributed to decline of the *P. veronii* mutant metapopulation. Grazing has frequently been proposed as a reason for poor inoculant survival (63–65), and indeed, our metatranscriptomic data on both *P. veronii*-inoculated and noninoculated sand and silt suggested up to 12-fold higher abundances of eukaryote reads attributable to bacterivorous groups (e.g., amoeba, ciliates, myxococci).

Despite these differences in the *P. veronii* mutant metapopulation behavior in the two soils, time analysis identified several genes with shared fitness effects. As expected

in terms of toluene metabolism, insertions in the *ipb*-genes coding for the first steps of toluene transformation were counterselected. This effect was less for genes coding for the *meta*-cleavage pathways of (methyl)catechols (i.e., *ipbC-H* and *dmp*). Both *ipb* and *dmp meta*-cleavage gene clusters are induced in the presence of toluene (52) but are seemingly redundant as judged from their observed minor fitness loss. Mutations in genes coding for a toluene efflux pump (*ttgGHI*) also had a fitness cost on *P. veronii*, similar as shown for other toluene-degrading pseudomonads (29, 66, 67). Interestingly, insertion mutants in the *ttgV* transcriptional regulator of the *ttgGHI* operon were positively selected during liquid suspended growth with toluene. The reason for this positive selection may be a higher inherent tolerance to toluene, like what has been observed for *Pseudomonas putida* DOT-T1E mutants with a *ttgV* deletion (67, 68). This may have been particularly important during growth on toluene in liquid, where it is more bioavailable than in soils (69).

As we expected from our experimental design (e.g., comparing proliferation in soils to that in liquid medium with the same added carbon substrate), the gene lists and associated metabolic pathways or biological components with fitness defects largely overlapped between soil and liquid. Perhaps ironically, the extent of detected fitness defects was twice as large in liquid than soil incubations, which suggests that, *au fond*, the soil environment is more promiscuous for growth. The reason for this may be a higher availability of different nutrients from which *P. veronii* can profit. The types of general fitness effects observed in both liquid and soils covered a wide range of metabolic pathways and routes for biosynthesis of lipids, amino acids, cofactors, or nucleotides, for example. This also indicates that such pathways are redundant and that their defects caused by single insertions (as expected in each individual *P. veronii* mutant) only become detectable after prolonged growth. It is also conceivable that mutants in such pathways are salvaged to some extent by uptake of metabolites excreted from nonaffected mutants in the same pool (like in the liquid suspension controls where mutant libraries are incubated *en masse*) or from resident bacteria in the soil.

Among the narrow list of soil-conditional genes, we did not find any major specific common enriched metabolic pathways or biological systems, except several nutrient transporters. Soil incubations further counterselected specifically for insertions in a set of regulatory genes and cell wall functions in sand and other nutrient transporters, cell wall- and lipopolysaccharide-modifying enzymes, and transposases in silt. This gives little evidence to support any interpretation of factors that would contribute to *P. veronii* competitiveness among a higher background of resident microbiota in the silt. We attempted to potentially improve the sensitivity of detecting fitness effects through summation of insertion counts across coordinated biological systems, such as type III or type VI secretion systems, but this also did not point to any obvious further differences between liquid and soil, or between silt and sand. The absence of a specific "soil" gene set in *P. veronii*, in contrast to what was previously seen with *S. wittichii* (48), can be interpreted as a generalist ecological strategy to maintain central carbon metabolism rather than environment-dependent metabolic reprogramming. This conclusion is further supported by the absence of a clear soil-specific *P. veronii* global transcriptomic reaction (54).

Finally, as had been observed before for Tn-seq-generated mutants of *S. wittichii* growing in soil (48), some insertions caused massive *P. veronii* fitness gain. This was particularly evident from the fitness increase in soil of mutants with insertions in almost all the genes for flagellar biosynthesis and assembly (insertions in three independent genome locations). This is curious because previous transcriptomic data had indicated that *P. veronii* flagellar genes are higher expressed during exponential growth in soils (54), suggesting motility as an important adaptive feature for soil establishment. Indeed, other studies have reported that flagella provide a fitness advantage to cells in soils through improved dispersal and access to nutrient resources (63, 70). This is supported by studies on *S. wittichii* RW1 showing that interruption in flagellar genes resulted in strong fitness loss in soil (48, 71), and on *Pseudomonas fluorescens*, demonstrating significant better survival of

motile than nonmotile strains 21 days after inoculation in nonsterile sandy loam (72). In contrast, flagella biosynthesis and motility itself are costly, and mutants with synthesis defects might be selected in soils under conditions of constant humidity and plenty of carbon availability as in our experimental designs. Indeed, our data indicated a 15% shorter generation time of the *fleQ*-deletion mutant during growth in soil in toluene, which can explain why flagella synthesis mutants gain advantage. Despite shorter generation times, such mutants were not selected in liquid suspension, suggesting there may be additional reasons for their accumulation in soil. Flagella biosynthesis and motility may thus be a two-edged sword for cells. On the one hand, flagella enable chemotaxis for nutrient availability (73), particularly during short and infrequent wetting events in unsaturated soils (74), and escape from predators (63). On the other hand, flagella synthesis poses an important fitness cost and may favor attack by flagellotropic phages (75). More systematic studies to evaluate the actual impact of motility on predation during strain inoculation into soil might be interesting to be carried out (64).

The high-throughput scanning of insertion mutant behavior under environmentally relevant conditions thus provided us with a global picture of the functional repertoire for *P. veronii* to proliferate in contaminated soils. We must conclude that although many functions are important for proliferation in soil, most of these are not exclusive because they are also found in minimal medium suspended growth. Their fast adaptability and large redundant genomes enable pseudomonads to maintain their main central metabolic routes and grow fast despite changing environmental conditions. This seems different from the strategies of other bacteria (e.g., *Sphingomonas*), which mount a specific "soil-type" response and possibly grow more slowly. In addition to selecting strains for bioaugmentation based on their catabolic properties, future soil microbiome engineering efforts should thus take other ecological properties into account as well.

## MATERIALS AND METHODS

**Culture conditions.** *P. veronii* 1YdBTEX2 and *E. coli* strains were routinely grown in lysogeny broth (LB, 10 g L$^{-1}$ tryptone, 5 g L$^{-1}$ yeast extract, 10 g L$^{-1}$ sodium chloride) on a rotatory shaker at 180 rpm at 30°C or 37°C, respectively. As defined minimal medium (MM), we used type 21C mineral medium (76). Whenever necessary, the medium was solidified with 1.5% agar. As carbon sources for *P. veronii* we used 20 mM sodium succinate or toluene. Kanamycin was supplemented at 50 $\mu$g mL$^{-1}$ (Km50) to select for transposon insertions in *P. veronii* or for maintenance of plasmid pRL27 (see below). Toluene as a sole carbon source was supplied to cells growing on MM-agar plates through the vapor phase in a gas-tight reservoir.

**Preparation of phage lysate.** To obtain bacteriophage lysate for the transposon mutant library enrichment, 1 mL of Sextaphag polyvalent pyobacteriophage preparation (Microgen, Russia) (77) was added to 50 mL of *E. coli* BW20767 culture grown in LB with Km50 at a culture turbidity at 600 nm of 0.8 and incubated for 3 h at 37°C and 180 rpm. After that, the lysate was amended with 100 $\mu$L of 20 mg mL$^{-1}$ DNase I and incubated for 15 min as above to reduce viscosity. Finally, the lysate ($>10^9$ phage per mL) was cleared by centrifugation for 10 min at 17,400 $\times$ *g* in an FA-45-6-30 rotor (Eppendorf), filtered through 0.45 $\mu$m cellulose acetate and stored at 4°C until needed for enrichment (see below).

**Transposon mutant library construction.** Transposon mutant libraries of *P. veronii* were prepared through conjugation of a suicide plasmid from the *E. coli* BW20767 (pRL27) donor, which expresses a hyperactive transposase leading to insertion of a Km-resistant gene flanked by transposon ends (56). *P. veronii* and *E. coli* cultures were grown overnight in 80 mL of LB and LB-Km50, respectively, harvested by centrifugation at 3,220 $\times$ *g* for 7 min in an Eppendorf swing-out A-4-62 rotor at 25°C, and resuspended each in 40 mL of fresh LB for 1 h at 30°C without shaking to resume growth and remove traces of Km. Two culture aliquots, each with 1.8 $\times$ 10$^{10}$ donor cells and 2.8 $\times$ 10$^{10}$ recipient cells were then combined and centrifuged as described above. After removing the supernatant, the cell slurries were gently aspirated with a pipette, and each transferred to a 0.2-$\mu$m pore size 25-mm cellulose acetate filter disc (Sartorius). Disks were placed on LB-agar and incubated for 48 h at 30°C. The same volumes of donor and recipient cells were separately treated in the same way to serve as controls. Following the incubation, cells from each filter were resuspended in 2 mL phosphate-buffered saline (PBS) by vortexing, after which the two duplicate mating mixtures were pooled. Aliquots of 0.1 mL from all cell suspensions were serially diluted in MM and plated on selective media as described below. The remaining mating mixture volume (1.9 mL) was supplemented with glycerol to a final concentration of 15% (vol/vol), aliquoted, flash frozen in liquid nitrogen, and stored at −80°C until library enrichments.

*E. coli* BW20767 donor cells were counted on LB-Km50 agar plates incubated at 37°C, which is a non-permissive temperature for *P. veronii*. Wild-type *P. veronii* (recipient) was counted on MM agar plates incubated at 30°C in toluene vapor. Separately filter-incubated donor or recipient cells plated on selective media showed the absence of spontaneous appearance of mutants with Km resistance. The number

of potential *P. veronii* transposon insertion mutants was estimated by counting on MM-Km50 agar plates incubated with toluene vapor as the sole carbon source.

**P. veronii Tn5-mutant library enrichment.** To reduce the incidence of wild-type *P. veronii* and *E. coli* cells in the raw transposon libraries, we thawed an aliquot of 0.6 mL of the frozen mating mix (Lib1; with an estimated $3.6 \times 10^5$ Km-resistant *P. veronii* mutants) on ice and inoculated this into 50 mL of LB and 50 $\mu$g mL$^{-1}$ Km in a 250-mL-sized conical flask. The resulting liquid culture was amended with 2 mL of the *E. coli* BW20767-specific bacteriophage preparation and incubated for 48 h at 30°C and 180 rpm. Following the enrichment, we counted the number of Km-resistant *P. veronii* cells and the remaining *E. coli*, as described above. Subsequently, 100 *P. veronii* colonies grown on MM with toluene were verified for Km resistance by replica plating on nutrient agar (Oxoid) with Km, and 22 colonies were screened by PCR-amplification of a fragment of the inserted aminoglycoside phosphotransferase (*aph*) gene and a *P. veronii* genome marker using primer pairs aph_rev/aph_fw and Pv_chr2_fw/Pv_chr2_rev, respectively (57), which confirmed the presence of the insertion.

The whole procedure starting from the mating was repeated once more for the preparation of the Lib2 enriched library (estimated total $6.5 \times 10^7$ mutants). Lib2 was enriched in 50 mL MM with 20 mM succinate plus Km and 5 mL of phage lysate, which was incubated for 16 h at 30°C and 180 rpm rotary shaking. Both Lib1 and Lib2 enrichments were aliquoted and stored at −80°C with 15% glycerol. At the point of freezing, the stocks had approximately $3.6 \times 10^8$ Km-resistant *P. veronii* cells (estimated from CFU on MM-toluene-Km plates), and less than $1.9 \times 10^4$ *E. coli* (estimated from CFU on LB-Km at 37°C). The level of *E. coli* contamination was further estimated from the proportion of reads mapping to the full transposon (assuming this corresponded to pRL27; Table S1).

**Selection conditions in liquid medium.** To identify *P. veronii* mutants with insertions in genes causing fitness loss, we imposed the following selective conditions: (i) growth in MM with toluene as the sole carbon source, (ii) growth in sand microcosms, or (iii) in silt microcosms, both with toluene as added carbon source. For conditions (i) and (ii) we used the Lib1 library; for condition (i) and (iii) we used Lib2.

To prepare the libraries for inoculation in the selective conditions, 3.6 mL (Lib1) or 10 mL (Lib2) of the library glycerol stock was revived in 30 mL of LB with Km50 (Lib1) or in 100 mL of MM-Km50 (Lib2), incubated at 30°C and 180 rpm for 4 h until a culture turbidity of 0.42 (ca $5 \times 10^8$ cells mL$^{-1}$). An aliquot of 10-mL culture was centrifuged at $3,220 \times g$ for 8 min in an Eppendorf swing-out A-4-62 rotor at 25°C, after which the cell pellet was resuspended in 12 mL of MM without added carbon substrate. This cell suspension was further diluted 20-fold in the same medium to produce the inoculum for the selection experiments (ca $1 \times 10^7$ cells mL$^{-1}$). Four aliquots of 2 mL (of the washed but undiluted cell suspension with an optical density [OD$_{600}$] of ~1.1) were harvested by centrifugation as before, and cell pellets were stored at −80°C until DNA extraction. These four samples served as the $t = 0$ samples of Lib1 and Lib2.

For growth in liquid culture with toluene, we transferred 5 mL of the inoculum directly to 50-mL polypropylene tubes (Greiner AG, cat. no. 227261) in four replicates. A 1-mL pipette tip sealed on one end containing 0.2 mL pure toluene was placed into the tubes, after which they were closed with a screw cap and incubated upright at 25°C for 72 h. After incubation, the toluene-containing tip was removed from the tube and the cell suspension was vigorously vortexed, after which a 0.1-mL subsample was taken, serially diluted, and plated on MM with Km50 incubated in the presence of toluene to count the number of Km-resistant *P. veronii*. A further subsample of 0.1 mL of each of the four replicates was diluted in 5 mL MM into a new 50-mL Greiner tube, and again incubated with toluene for 3 d. This passaging was repeated 8 times in total for Lib1 and 3 times for Lib2, each corresponding approximately 6 to 7 generations of growth. The remaining culture volumes (4.8 mL) after cycles 1, 4, and 8 (Lib1) and 1, 2, and 3 (Lib2) were centrifuged as above to pellet the cells, which were frozen at −80°C until DNA isolation.

**Selection conditions in soil microcosms.** For growth in soil microcosms, we weighed 50 g of non-sterile air-dried sand or silt in 250-mL Schott flasks. Sand and silt were sampled at locations previously described in (54), air-dried for 7 to 10 days at room temperature, and sieved to remove >2-mm particles before being filled in the flasks. Four replicate microcosms were started simultaneously, which were inoculated with 5 mL of the reconstituted and diluted library suspension mentioned above. Lib1 was used for the sand and Lib2 for the silt microcosms. After inoculation, the material was mixed manually with a sterilized spatula and placed on a roller board (IKA roller 6 digital, set to 30 rpm) for 30 min. Flasks were then placed upright and a 1-mL pipette tip with one end sealed was placed inside with 0.4 mL pure toluene. The soil microcosms were closed with a Teflon-lined screw cap and incubated without shaking at 25°C for 3 days. After incubation, the tip was removed from the flask and the content of the microcosm was homogenated by rotation on the roller board during 30 min. An aliquot of 5.5 g was retrieved and used to inoculate the next soil microcosm with 45 g dried material, to which 4.5 mL MM was added to maintain the same gravimetric water content. This passaging was repeated 15 times. Each cycle was considered to allow ~3 generations of growth in soil. The remaining material was used for counting *P. veronii* and for DNA isolation (see below).

**P. veronii mutant constructions.** One gene in the flagellar gene region was targeted for specific deletion in the *P. veronii* chromosome 1 (PVE_r1g4308, *fleQ* homolog). Then, 800-bp regions flanking the gene were PCR-amplified and cloned into EcoRI-SalI digested vector pEMG (78) using the ClonExpress Ultra one-step cloning kit (Vazyme, no. C115). PCR primers were designed to leave only the first six and last six amino acid codons. After Sanger sequence verification, the resulting genetic constructs were mobilized via triparental mating from *E. coli* DH5$\alpha$ $\lambda$-pir with the help of mobilization helper pRK2013 (79) into *P. veronii*. Gene deletion by recombination was forced by ISceI nuclease digestion, as described (78). Putative deletion mutants were verified by PCR and sequencing of the missing locus.

**Counting *P. veronii* wild type or mutants in soil microcosms.** For counting, aliquots of 11 g microcosm material were transferred into 50-mL Greiner tubes, amended with 9 mL of MM, after which the

tube was vortexed at maximum setting for 1 min (Vortex Genie 2, Scientific Industries Inc.). Larger soil particles were allowed to sediment for a few minutes, and the supernatant was transferred to a fresh 50-mL tube. An aliquot of 50 $\mu$L was removed and serially diluted in MM, which was plated on MM-Km50 and incubated with toluene vapor to count the number of CFU of Km-resistant toluene-degrading *P. veronii*. The remaining volume was centrifuged for 8 min at 3,220 × *g* to pellet the cells, which were frozen with liquid nitrogen and stored at −80°C for DNA extraction and library preparation (see below).

*P. veronii* wild type and *fleQ* mutant were grown individually in replicate soil microcosms ($n = 4$), containing 100 g nonsterile silt in a 500-mL Schott screw cap glass bottle. Strains for inoculation were grown at 30°C and 180 rpm in liquid MM supplemented with toluene vapor until turbidity ($OD_{600}$) of 1, then diluted 10,000-fold in MM without a carbon source. Next, 10 mL of this inoculum was added to the soil microcosm, mixed on a roller board as before, and incubated in an upright position at 25°C in the presence of toluene vapor (0.2 mL in a sealed 1-mL pipet tip). At defined time points, soil samples (11 g) were removed from the microcosms, extracted with 9 mL MM by 2-min vortexing, serially diluted and spot plated (5 $\mu$L) on MM-agar incubated with toluene vapor in a jar. Colonies were counted using a binocular after a 3-d incubation.

The same inoculants were 40-fold diluted in MM containing 10 mM succinate and 200-$\mu$L portions were aliqoted in six replicates into a polystyrene 96-well microtiter plate (Starlabs CC7672-7596). Growth was monitored at 10-min reading intervals by measuring OD at 600 nm ($OD_{600}$) in a Biotek Synergy H1 plate reader (Agilent) set to 30°C and with 425 rpm dual orbital shaking.

**DNA extraction and quantification.** Total DNA was extracted from frozen cell pellets using the DNeasy PowerSoil kit (Qiagen, cat. no. 12888-100) according to the manufacturer's recommendations. DNA quantity and quality in the purified solutions were assessed using a Qubit 3.0 dsDNA BR assay kit (ThermoFisher Scientific) and by agarose gel electrophoresis.

**DNA library preparation and Illumina sequencing.** Aliquots of 0.5 $\mu$g DNA were used for library preparation using the NEBNext Ultra II FS CBA library prep kit for Illumina (New England Biolabs) in a customized protocol. First, the DNA sample was enzymatically fragmented (10 min), end repaired, and dA-tailed, followed by ligation to NEBNext loop adaptors and USER enzyme treatment (according to supplier's instructions). Excess adapters and small (<100 bp) fragments were removed using 0.6× CleanNGS (CleanNA, CNGS-0050) SPRI magnetic bead purification. Purified adapter-ligated fragments containing transposon insertions were subsequently enriched by two steps of PCR amplification. The first PCR (21 cycles) combined a customized primer complementary to the left-end transposon end (TnL) and one of 24 NEBNext multiplex oligonucleotides for Illumina (index primer set 1 and set 2 – New England Biolabs. Primer sequences specified in [57]). The resulting amplicons were size selected (300 to 700 bp) using two-sided SPRI-bead purification with 0.7 to 0.5 left-right volume ratios. In a final 6-cycle PCR, all individual indexed libraries containing the junctions between the TnL–end and *P. veronii* genome targets were amplified with one of four nested TnL-anchored primers (TnL_uni_xxx_P5, [57]), each containing the P5-adapter sequence for Illumina and a phasing box, in combination with the TruSeqP7_Fw primer (57). The resulting DNA amplicons were purified twice using 0.8× vol of SPRI beads and two 80% ethanol washes. DNA concentrations in the prepared libraries were measured using Qubit assays as above, then pooled by groups in equal proportions (40 ng), which were analyzed on an Agilent 2100 bioanalyser (Agilent Technologies) to confirm quality and DNA size distribution. Pooled libraries were sequenced in-house on a MiniSeq instrument (Illumina, Inc., USA) using the MiniSeq system mid output kit 150 cycle (Illumina, Inc., USA), or on a MiSeq instrument with a 150-cycle MiSeq reagent kit v3 with custom sequencing primers (TruSeq Read1 and TruSeq_IndexRead1, [57]), at the Lausanne Genomics Technology Facilities. All replicate raw data sets are accessible from the Short Read Archives under BioProject ID PRJNA741928.

**Processing of sequencing data.** Depending on the sample, we obtained between 284,321 and 2,585,631 raw sequence reads (151 bp, Table S1). Reads without the 19-bp TnL-end sequence were removed by using Cutadapt (version 2.5). Next, we removed the Illumina adaptors using Trimmomatic (version 0.36; default parameters) (80). With a custom-made in-house *Perl* script we removed all the remaining sequences that contained only transposon sequence and no connecting genome insert position sequence. The filtered, trimmed, and cleaned fastq files were then quality checked using FastQC version 0.11.5. The remaining quality controlled reads were aligned and mapped to the *P. veronii* genome (European Nucleotide Archive under bioproject number PRJEB11417) using bowtie2 under standard stringency (version 2.3.4.1) (81). Finally, the mapped reads were grouped per gene or intergenic region by an in-house *Perl* script, and individual insertion positions were counted from the *sam* files using UNIX commands. Insertions mapping in intergenic regions were counted, removed, and not further considered for analysis (Table S1).

**Statistical analysis.** To estimate the deviation between the observed per-gene insertion distributions in the two starting libraries (Lib1 and Lib2) and expected random insertion distributions without selective effects, we four times independently randomly subsampled $1.2 \times 10^5$ unique insertions (close to the observed number) without replacement from the *P. veronii* combined genome (chromosome 1 and 2, and plasmid). These were then grouped per gene to have four replicates of expected random distributions, normalized by total, and compared to the sum-normalized Lib1 and Lib2 per-gene distributions using a *mattest* and *mafdr* in MATLAB (Mathworks, versus R2021b) to estimate significant differences. Essential genes were considered those if their ratio of mean-observed to mean-expected distributions was <0.05, and then overlapped between both starting libraries. The KEGG-pathway and COG attributions of the resulting gene list were compared to attributed gene lists of the same size randomly subsampled from the full genome and displayed using iPath3 (82).

To identify genes whose insertions influence *P. veronii* fitness in soil, we removed extreme outliers (top 0.1% percentile and those surpassing 100,000 reads in the respective sample; Table S2), and normalized the resulting insertion counts using the "Pseudoreference Sample" model (83), taking into account the median read density across genes per sample and per library origin (i.e., Lib1 and Lib2 samples normalized among each other). These were then further normalized to the same total read sum per sample by data sampling from a probability distribution function produced from the observed read counts. The resulting data set was then used to calculate the ratio of the mean gene insertion count at start and the collective mean insertion count over the three time samples $T_0/T_{1-3}$, and to estimate the *P* value and its corresponding false-discovery rate (FDR, *mattest* and *mafdr* in MATLAB). Insertion counts were then considered to have increased over time (*up*) if the ratio $T_0/T_{1-3}$ was <0.5 and the FDR was <0.05. Conversely, counts were considered *down* (depleting over time) at a ratio $T_0/T_{1-3}$ >0.5 and FDR <0.05. Any previously considered essential gene was excluded from the *up* and *down* gene lists.

Gene lists of enriched (*up*) or depleted (*down*) insertions were attributed to COG or KEGG categories based on the *P. veronii* annotation (55), and their proportional attribution was compared to expected attributions from the full genome by $n = 100$ random subsamples of the same size as the respective gene list. Confidence interval limits ($P = 0.01$) on the mean of the random subsamples were calculated by bootstrapping ($n = 1,000$). *P* values and false discovery rates were determined from the mean and standard deviations of the random subsamples (assuming normal distribution, *normpdf*; and BHFDR = true in MATLAB 2021b). To identify gene insertions with either specific liquid or soil growth effects, we derived the common as well as unique elements in the *up* and *down* gene lists established above.

To find the overlap of fitness-affected *P. veronii* genes to the essential gene set of *Acinetobacter baylyi* ADP1 (42), we compared gene sets in pairwise BLASTp using thresholds of e value <1e-10, and percent identity of >40.

All scripts, further methods, and raw and processed analyzed data are provided as a single downloadable item from the accompanying link at Zenodo [57].

**Diversity analysis.** Both soils (silt and sand) were characterized for bacterial taxonomic diversity using amplicon analysis of the V3-V4 region of the 16S rRNA gene on purified DNA, as described (84). Raw reads are available from a single downloadable link [57].

Eukaryotic diversity was assessed from metatranscriptomic data sets of silt or sand inoculated with *P. veronii* or not for 1 h, in the presence of toluene vapor ($n = 4$ replicate per treatment), mapped by MGNify (85) on the European Nucleotide Archive (accession numbers, ERS2210331 to ERS2210346). These experiments were previously outlined for the global transcriptome of *P. veronii* in different soils and conditions (54). As potential bacteriovores, we considered protozoan zooflagellates, small flagellates, gliding bacteriovores, myxomycetes, and amoebozoa. Normalized read proportions of the 18S rRNA gene corresponding to eukaryotic taxa of the above-mentioned classes were summed, and sums were compared among the four replicates for differences between silt and sand, in absence or presence of inoculated *P. veronii* (Table S4).

## SUPPLEMENTAL MATERIAL

Supplemental material is available online only.

**FIG S1**, PDF file, 3.7 MB.

**FIG S2**, PDF file, 0.9 MB.

**TABLE S1**, XLSX file, 0.02 MB.

**TABLE S2**, XLSX file, 0.1 MB.

**TABLE S3**, XLSX file, 0.02 MB.

**TABLE S4**, XLSX file, 0.01 MB.

**TABLE S5**, XLSX file, 0.01 MB.

**TABLE S6**, XLSX file, 0.02 MB.

## ACKNOWLEDGMENTS

The authors acknowledge the help of Noushin Hadadi in initial analysis of gene fitness data sets.

This work was supported by the MicroScapesX grant from the Swiss Initiative in Systems Biology SystemsX.ch and by the National Centre of Competence Research in Microbiomes.

M.M., D.V., N.C., S.C., and V.S. carried out experimental work. M.M. and J.M. conducted data analysis. M.M. and J.M. wrote the manuscript. M.M., D.V., N.C., S.C., V.S., and J.M. commented on the final text. J.M. raised funding.

We declare no conflict of interest.

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
