## [Reviewer comments · mSystems]

Fitness-conditional genes for soil adaptation in the bioaugmentation agent *Pseudomonas veronii* 1YdBTEX2

Marian Morales, Vladimir Sentchilo, Nicolas Carraro, Senka Causevic, Dominique Vuarambon, and Jan van der Meer

Corresponding Author(s): Jan van der Meer, Universite de Lausanne

Review Timeline:

Submission Date:

November 28, 2022

Accepted:

January 10, 2023

Editor: Laura Hug

Reviewer(s): Disclosure of reviewer identity is with reference to reviewer comments included in decision letter(s). The following individuals involved in review of your submission have agreed to reveal their identity: Trevor C Charles (Reviewer #1)

Transaction Report:

DOI: <https://doi.org/10.1128/msystems.01174-22>

January 10, 2023

Prof. Jan Roelof van der Meer
Universite de Lausanne
Department of Fundamental Microbiology
Batiment Biophore
Quartier Unil-Sorge
Lausanne CH 1015
Switzerland

Re: mSystems01174-22 (Fitness-conditional genes for soil adaptation in the bioaugmentation agent *Pseudomonas veronii* 1YdBTEX2)

Dear Prof. Jan Roelof van der Meer:

Your manuscript has been accepted, and I am forwarding it to the ASM Journals Department for publication. For your reference, ASM Journals' address is given below. Before it can be scheduled for publication, your manuscript will be checked by the mSystems production staff to make sure that all elements meet the technical requirements for publication. They will contact you if anything needs to be revised before copyediting and production can begin. Otherwise, you will be notified when your proofs are ready to be viewed.

If you would like to submit a potential Featured Image, please email a file and a short legend to msystems@asmusa.org. Please note that we can only consider images that (i) the authors created or own and (ii) have not been previously published. By submitting, you agree that the image can be used under the same terms as the published article. File requirements: square dimensions (4" x 4"), 300 dpi resolution, RGB colorspace, TIF file format.

We recognize that the video files can become quite large, and so to avoid quality loss ASM suggests sending the video file via <https://www.wetransfer.com/>. When you have a final version of the video and the still ready to share, please send it to mSystems staff at msystems@asmusa.org.

Sincerely,

Laura Hug
Editor, mSystems

Journals Department
E-mail: mSystems@asmusa.org